# Comparative Study of Tribological Behavior of TiN Hard Coatings Deposited by Various PVD Deposition Techniques

Peter Panjan [1,*], Aljaž Drnovšek [1], Pal Terek [2], Aleksandar Miletić [2], Miha Čekada [1] and Matjaž Panjan [1]

1   Jozef Stefan Institute, Jamova 30, 1000 Ljubljana, Slovenia; aljaz.drnovsek@ijs.si (A.D.); miha.cekada@ijs.si (M.Č.); matjaz.panjan@ijs.si (M.P.)
2   Faculty of Technical Sciences, University of Novi Sad, 21000 Novi Sad, Serbia; palterek@uns.ac.rs (P.T.); miletic@uns.ac.rs (A.M.)
*   Correspondence: peter.panjan@ijs.si

**Abstract:** In this paper, we present a comparative study of tribological properties of TiN coatings deposited by low-voltage electron beam evaporation, magnetron sputtering and cathodic arc deposition. The correlation of tribological behavior of these coatings with their intrinsic properties and friction condition was studied. The influence of surface topography and the surrounding atmosphere was analyzed in more detail. We limited ourselves to the investigation of tribological processes that take place in the initial phase of the sliding test (the first 1000 cycles). A significant difference in the initial phase of the sliding test of three types of TiN coatings was observed. We found that nodular defects on the coating surface have an important role in this stage of the sliding test. The tribological response of TiN coatings, prepared by cathodic arc deposition, is also affected by the metal droplets on the coating surface, as well as those incorporated in the coating itself. Namely, the soft metal droplets increase the adhesion component of friction. The wear rates increased with the surface roughness of TiN coatings, the most for coatings prepared by cathodic arc deposition. The influences of post-polishing of the coating and the surrounding atmosphere were also investigated. The sliding tests on different types of TiN coatings were conducted in ambient air, oxygen and nitrogen. While oxygen promotes tribo-chemical reactions at the contact surface of the coating, nitrogen suppresses them. We found that the wear rate measured in ambient air, compared with that in an oxygen atmosphere, was lower. The difference is probably due to the influence of humidity in the ambient air. On the other hand, wear rates measured in a nitrogen atmosphere were much lower in comparison with those measured in an oxygen or ambient air atmosphere.

**Keywords:** TiN hard coating; low-voltage electron beam evaporation; unbalanced magnetron sputtering; cathodic arc deposition; surface topography; tribology

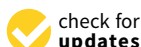



## 1. Introduction

Hard protective coatings started to be used on an industrial scale in 1969, when Sandvik Coromant and Krupp-Widia companies introduced chemical vapor deposition (CVD) technique for deposition of TiC coatings almost at the same time [1]. The CVD titanium nitride (TiN) coating was developed soon after TiC and reached the commercial exploitation stage in the early 1970s [2]. The high temperature (typically about 1000 °C) limited the application of such coatings mainly to cemented carbide tools, but they were not suitable for wear protection of high speed steel (HSS) tools. This shortcoming of CVD processes was overcome a decade later by the introduction of physical vapor deposition (PVD) processes [3]. PVD techniques enabled deposition of TiN hard coatings at temperatures of less than 500 °C, which is below the tempering temperature for most types of tool steels. The commercial exploitation of the first PVD TiN hard coatings started in the late 1970s, and their use underwent a remarkable boom especially in the field of metal machining. Experts in the field of metal machining believe that the protection of

cutting tools with wear-resistant TiN hard coatings has been one of the most significant technological advances in the development of modern tools. In the early years of PVD hard coatings, titanium nitride was just about the only coating. Although many other PVD hard coatings were later introduced in the market, the TiN coating is still today one of the most important.

The first PVD TiN hard coatings were deposited by *low-voltage electron beam evaporation* (Balzers) [3]. This technology was commercially very successful, and TiN coatings prepared using this technique are still a quality standard in this field. Almost at the same time, the USA Multi-Arc company started producing TiN hard coatings by *cathodic arc deposition* [4]. Their technology was based on a license bought in the former Soviet Union [5]. In the review paper [6], Anders reported that much of the early work on cathodic arc technology was performed in the former Soviet Union with the first industrial systems in 1974 (*Bulat technology*).

Initially, the TiN coatings prepared by the conventional DC magnetron sputtering technique were significantly inferior to those prepared by other PVD techniques. This process did not provide a sufficient intensity of ion bombardment of the substrate surface during the deposition process. Therefore, sputter-deposited coatings were usually not fully dense. In the second half of the 1980s, this problem was overcome with the invention of *the closed-field unbalanced magnetron sputtering* technique by Window and Savvides [7], which provided a degree of substrate ion bombardment during the deposition process equivalent to that of competing PVD hard coating techniques. An even greater advance in this field was made about 20 years later with the introduction of *high-power impulse magnetron sputtering* (HIPIMS) deposition, which provides an extremely high ionization fraction of the sputtered species [8]. All of these improvements have enabled the preparation of high-quality coatings (higher density, improved adhesion, reduced roughness, and more uniform coating of high aspect ratio features) by magnetron sputtering.

The chemical bonds in TiN material are a combination of covalent, metallic and ionic bonds. This results in a unique combination of properties that are typical for metals (good electrical and thermal conductivity) as well as those that are typical for ceramics (high hardness, chemical inertness and high melting temperature). Due to such an unusual combination of properties, TiN coatings have found widespread application in industrial production.

TiN is a simple and cost-effective coating for *wear protection* of cutting, plastic molding and cold forming tools. It is suited for wear protection of HSS cutting tools working at low and medium cutting speeds. Typical applications include drilling, milling and turning of mild steels at cutting speeds below 100 m/min. Due to its gold color, TiN coating is also an excellent indicator for wear. The wear can be clearly seen on the worn areas of the tool or components after a certain time in service. However, the TiN coating is not suitable for the protection of tools used for machining difficult-to-cut workpiece materials (hard or "sticky") or for high-speed machining due to low oxidation resistance (below 600 °C). The surface oxidation of a TiN coating has a great impact on its wear resistance. The oxide layer formed on TiN coating easily spalls off due to high compressive stresses that develop because of the large difference in the molar mass ratio of oxide compared to nitride.

The unique gold color, abrasion resistance, and chemical inertness of TiN coatings have also encouraged their widespread applications for *decorative purposes* [9–11]. Watchmakers were the first group interested in the decorative aspect of TiN hard coatings. It is especially important that the preparation of PVD decorative coatings, unlike electrochemical ones, is not environmentally hazardous. The color of solid materials originates from electronic band structure or density of states. The golden yellow color of the pure and stoichiometric TiN coating is the consequence of the reflection edge located in the visible region with a characteristic reflectivity minimum of about 450 nm [12]. The location of the reflection edge can be affected by variations in composition, formation of lattice defects, or incorporation of impurities (e.g., oxygen, carbon). However, the color and spectral reflectivity of TiN films can also depend on surface roughness, arising from the columnar growth and formation of

growth defects. Today, decorative applications of TiN coatings include sanitary hardware, household appliances, door handles, jewelry luxury items and architectural glass (to obtain special optical functions such as IR reflection). The use of decorative coatings often also allows the replacement of substrates made of expensive materials with cheaper ones.

TiN is non-toxic, and for this reason, it is used in a number of *biomedical applications* [13–15]. Because of its intrinsic biocompatibility, antibacterial properties, chemical inertness and good tribological properties, TiN coating is a suitable material for wear protection of orthopedic implants (hips, knees and other joints) and dental implants (screws, abutments) [14]. To protect the prosthetic joint, it is important that the coating material be hard and chemically inert to prevent wear, corrosion and the formation of debris that causes tissue inflammation. Due to hemocompatibility, TiN coating is used also in cardiology for ventricular assist devices for patients with heart failure and for pacemaker leads. For a long time, the medical industry has used PVD coatings to improve surgical tools (e.g., scalpels, blades, drills, reamers, orthopedic bone saw blades), where corrosion protection, sharpness and cutting edge retention are important (sharp cutting edges on medical instruments allow wounds to heal faster while the antimicrobial properties of TiN coatings reduce the possibility of infections) [13]. TiN coatings also improve surgical tool identification and reduce glare in the operating room.

Due to its relatively high thermal and structural stability combined with low electrical resistivity, TiN is employed in microelectronic and photovoltaic devices as an advanced metallization material or *diffusion barrier* [16]. In these devices, a broad variety of materials, ranging from metals and semiconductors to insulators, are in contact with each other. TiN diffusion barrier films prevent the intermixing and interdiffusion of these materials during the fabrication and operation of such devices and the resulting loss of their functionality. Its remarkable properties make titanium nitride film an attractive candidate also for other applications. These include gate electrodes in advanced field-effect transistors and emerging resistive switching memory technologies [17], cathodes for high energy density lithium-sulfur batteries [18] and electrodes in photovoltaic cells [19].

In each of these applications, with the exception of microelectronics, the good tribological properties of TiN coating are of crucial importance. In the past, a number of reports on the tribological performance of TiN coated surfaces have been published [20–22]. In the majority of these studies, researchers focused on the steady-state value of the friction coefficient, while the mechanism of friction and wear in the initial phase of the sliding test was largely ignored. Further, only a few papers have described the impact of coating surface roughness and surrounding atmosphere on the tribological properties of PVD hard coatings [23–25]. In this paper, we compared the tribological behavior of stoichiometric single-phase TiN coatings deposited by low-voltage electron beam evaporation, magnetron sputtering and cathodic arc deposition. All experiments were performed on samples prepared in production batches. We analyzed how the tribological properties of TiN coatings, prepared by these techniques, depend on the microstructure, texture and especially surface topography. The influences of post-polishing of the coating and the surrounding atmosphere were also investigated.

## 2. Materials and Methods

All TiN stoichiometric coatings were prepared in industrial batch-type deposition systems: a BAI730 low-voltage electron beam evaporation system (Blazers, Vaduz, Liechtenstein), CC800/7 modified unbalanced magnetron sputter deposition system (Cemecon, Würselen, Germany) and AIPocket cathodic arc deposition system (KCS Europe GmbH, Monschau, Germany). In the remainder of this paper, the abbreviations BAI, CC7 and AIP will be used for these three deposition techniques, as well as for their corresponding coatings. A more extensive description of all three deposition techniques can be found in a recently published paper [26], while the essential process parameters are given in Table 1. Different deposition techniques result in different physical and mechanical properties of TiN coatings.

**Table 1.** Deposition methods and process parameters used for the preparation of TiN hard coatings. The deposition rates and ion current density are averages since all deposition techniques exhibited both spatial and temporal variations during a given deposition run as well as from one run to another run (depending on batching material).

| - | - | BAI | CC7 | AIP |
|---|---|---|---|---|
| Preheating | heating method | electron bombardment | infrared heating | infrared heating |
| - | preheating temperature (°C) | 450 | 450 | 450 |
| Etching | etching mode | DC | RF | pulsed DC |
| - | working gas | Ar | Ar + Kr | Ar |
| - | negative substrate etching voltage (V) | 200 | 200 | 300/400 |
| - | etching time (min) | 15–30 | 80 | 30 |
| Deposition | deposition method | low voltage electron beam evaporation | unbalanced magnetron sputtering | cathodic arc evaporation |
| - | temperature (°C) | 450 | 450 | 450 |
| - | working gas | Ar + $N_2$ | Ar + Kr + $N_2$ | $N_2$ |
| - | pressure of working gas (Pa) | 0.2 | 0.7 | 4 |
| - | deposition time (min) | 80 | 125 | 45 |
| - | negative substrate bias voltage (V) | 125 | 120 | 70 |
| - | average deposition rate * (nm/s) | 0.85 | 0.36 | 1.56 |
| - | thickness (μm) | 4.1 | 2.7 | 4.2 |
| - | average substrate current density (mA/cm$^2$) | 3–5 | ~2 | - |

* for one-fold rotation of substrates.

Cold work tool steel AISI D2 (~58 HRC), produced by Ravne steel factory (Ravne na Koroškem, Slovenia), was used as a substrate material. Substrates were ground and polished to a mirror-like finish ($R_a$ < 12 nm). Before the deposition, substrates were degreased and cleaned in ultrasonic baths with detergents (alkaline cleaning agents, pH ~ 11), rinsed in deionized water, and dried in clean hot air. Prior to the coating deposition, substrates were heated up to the deposition temperature (450 °C) and sputter-etched in order to remove the native oxide and other contamination. One-fold rotation of the steel substrates was applied.

The crystal structure of as-deposited coatings was examined by X-ray diffraction (XRD) with CuK$_\alpha$ radiation using a Bruker diffractometer (AXS Endeavor D4, Billerica, MA, USA) in Bragg/Brentano mode and equipped with a CuK$_\alpha$ X-ray source (0.15406 nm). The spectra were collected using a scan step size of 0.02° in a diffraction angle between 20° and 100°.

The surface topography characterization of the coated and uncoated substrate was carried out using 3D stylus profilometry (Bruker Dektak XT, Billerica, MA, USA), and scanning electron microscopy (SEM, JEOL JSM-7600F, Tokyo, Japan). The microstructure and the coating morphology were studied using fracture cross-sections examined in a field emission scanning electron microscope (FEI Helios Nanolab 650i, Amsterdam, The Netherlands). Cross-sections were also prepared by focused ion beam techniques (FIB) using an FIB source integrated into the FEI SEM scanning electron microscope. SEM images were recorded using the ion beam and the electron beam.

Tribological properties were evaluated by ball-on-disk tribometer (CSM, Neuchatel, Switzerland) using a linear reciprocating mode. Tests were conducted at room temperature

in ambient air, nitrogen, and oxygen atmospheres. An alumina ball with a diameter of 6 mm was used as a counter-body. The normal load was 5 N, displacement amplitude 5 mm, number of cycles 1000. All coatings were tested both in the as-deposited and the post-polished surface conditions. The sliding tests were conducted at least three times on each sample, and the results were averaged. The coating wear track depth, the coating wear volume, and wear scar on the alumina ball were measured by a 3D stylus profilometer. Each of the measurements covered an area of 1 mm × 1 mm. The average cross-section area of a wear track was calculated from a series of profiles that composed the 3D profilometer image of the wear track. Wear rate was calculated as W = V/(F × L), where V is the wear volume (mm$^3$), F is the normal load (N), and L is the total sliding distance (m). Post-test characterization by scanning electron microscopy was also performed in order to investigate the mechanical and tribological response of the coatings. In order to detect chemical changes on worn surface, backscattered electrons (BSE) were recorded, since the atomic number contrast enables the identification of oxides.

The Vickers hardness of the coatings was measured with a nanoindenter (Fischerscope, Sindelfingen-Maichingen, Germany) using a load in the range of 25–1000 mN. The indents using the lower load were all located in flat areas free from visible defects, resulting in hardness values for the defect-free coating material.

### 2.1. Coating Microstructure

A comparison of SEM images from fracture-cross sections and FIB cross-sections of the BAI, CC7, and AIP TiN coatings deposited on D2 tool steel substrate are shown in Figure 1a–f, respectively. The SEM images demonstrate that all coatings, regardless of the deposition technique, exhibit columnar microstructure. Columns extend along the coating growth direction and are composed of grains oriented in the direction of the surface normal. The columnar morphology of the coating is mainly determined by the substrate surface roughness, preferential growth of crystal grains, and the surface mobility of the condensing atoms [27]. The columnar microstructure is a result of the growth competition between the adjacent grains. Due to a geometrical shadowing effect, columns growing faster prevent the growth of the slower ones. However, the microstructure is not homogeneous throughout the coating thickness, but changes due to the renucleation of new grains. Renucleation, which occurs during the growth at the sites of surface defects created by the impinging ions, disrupts the columnar microstructure. The newly formed grains are smaller and more equiaxed, although still elongated in the growth direction. This is particularly noticeable in the BAI coating, where the average column size is the smallest. The fracture cross-section SEM image (Figure 1a) of this coating shows that only a small fraction of the columns extends from the substrate to the top surface of the coating. A SEM image of the FIB cross-section recorded by ions (Figure 1d) shows that the BAI coating is composed of small grains (5–20 nm). On the other hand, the largest column size was observed in the AIP coating (with a diameter in the range of 0.5–1 μm), while the size of columns in CC7 coating were somewhere in between.

TiN coatings prepared by the three different deposition techniques, however, differ not only in the details of columnar microstructure but also in their coating porosity, degree of preferred orientation, and surface roughness. These differences in the properties of the coating are directly related to differences in the deposition processes and deposition parameters. While all the coatings were deposited on the same type of tool steel substrates (D2) and at comparable substrate temperatures (about 450 °C), the system geometries, coating growth rates, discharge conditions and the substrate pretreatment (ion etching) were quite different (see Table 1). The microstructure of TiN coatings primarily depends on the energy of the impinging atoms and ions (Ti$^+$, Ar$^+$, N$^+$), and the ratio ($j_i/j_{Ti}$) of the accelerated-ion flux $j_i$ to the flux $j_{Ti}$ of the deposited Ti atoms [28]. It is preferred that the ion flux reaching the substrate is high, while the ion energy is low (<20 eV) in order to avoid coating damage. A moderate ion bombardment during the coating growth increases adatom mobility and promotes the formation of nucleation sites and chemical reactivity. All

of these processes, together with recoil implantation and redeposition of depositing atoms, can disrupt the columnar microstructure (i.e., transition from columnar to equiaxed growth) and increase the coating density. Considering all these facts, the different microstructures of the BAI, CC7 and AIP coatings can be explained by the difference in the bias voltage (it was highest for the BAI coating and lowest for the AIP one) and also by the different ($j_i/j_{Ti}$) ratio, which was largest during deposition in the BAI system (around 2.5).

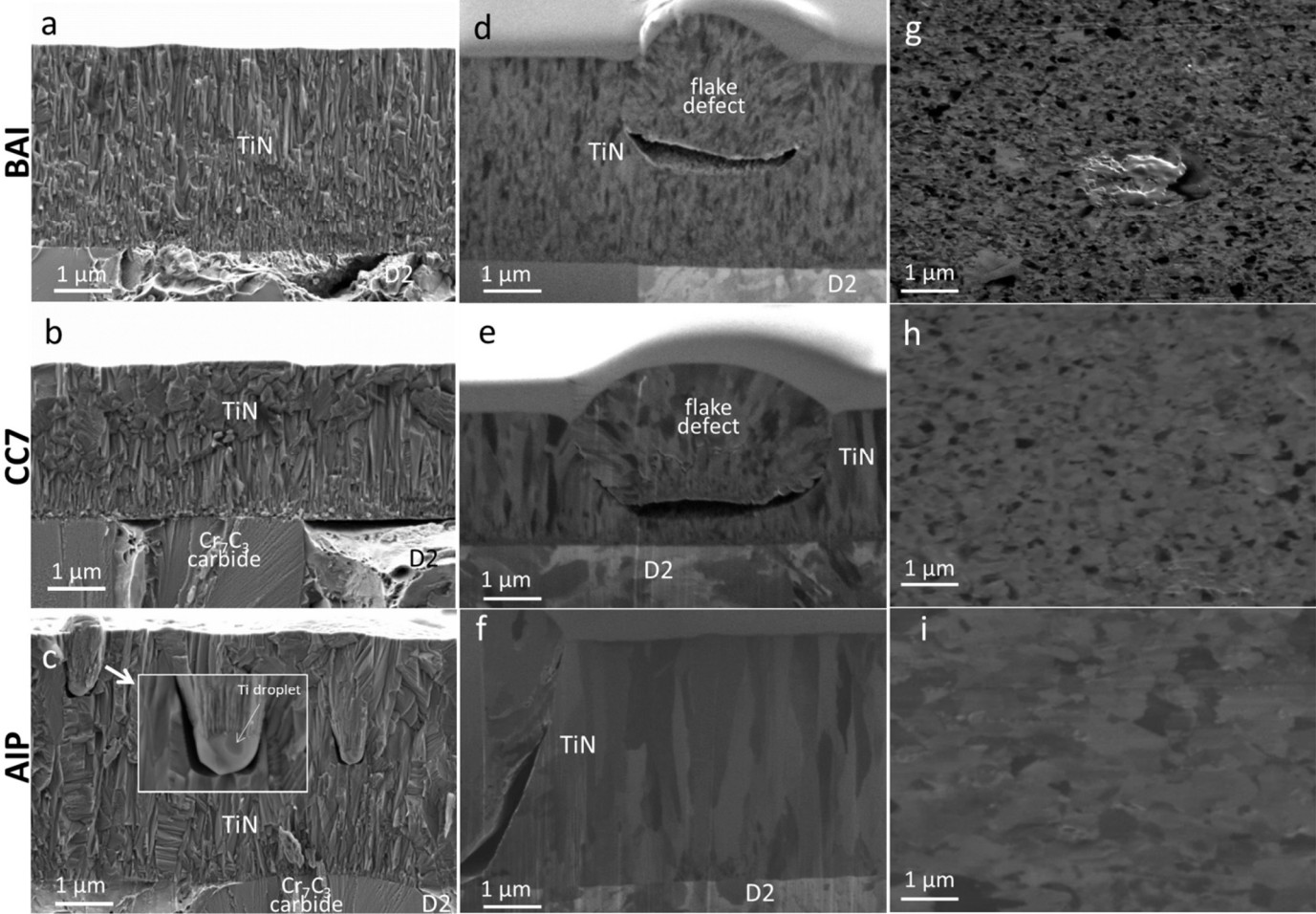

**Figure 1.** Fracture cross-sectional SEM images (**a**–**c**), FIB secondary electron images of cross-sections (**d**–**f**), and FIB secondary electron top view images (**g**–**i**) of TiN coatings deposited by low-voltage electron beam evaporation, magnetron sputtering and cathodic arc deposition techniques.

In order to obtain additional information about large area surface microstructural features, we used an innovative method based on top-view SEM images recorded using the ion beam. A pre-condition for performing an analysis is a very smooth coating surface, such as the one that formed in the wear track during a running-in period of a sliding test, where all protrusions were removed. The next step is a low-current ion etching with $Ga^+$ ions (no tilt applied on the sample stage) in the FIB module of the SEM microscope in order to remove the contamination layer from the coating surface. A clean and smooth coating surface is suitable for SEM imaging with an ion beam, in which the image contrast arises as a consequence of the ion channeling effect. When ions penetrate deeper into the crystal grain (the incident beam is parallel to a set of crystallographic planes inside a grain), the grain will appear darker due to a decrease in the number of secondary electrons that are emitted. However, if the grain has a no-channeling crystallographic orientation, it will appear bright. SEM imaging of coating surfaces with ions, therefore, provides information about grain preferential orientation. In order to produce a good quality image and to avoid

any artifacts that may form as a result of ion exposure, the ion beam current density and ion dose need to be small. SEM images of polished BAI, CC7 and AIP TiN coating surfaces recorded using the ion beam are shown in Figure 1g–i. The differences in the size of crystal grains and their preferred orientation are evident.

### 2.2. Coating Topography

In our recently published study [26], we described in more detail the topography of PVD hard coatings, prepared by different PVD deposition techniques on various substrate materials. We showed that the surface topography can be distinguished on several size scales. It originates from the topography of the substrate surface, intrinsic coating micro-topography and growth defects that form during the deposition process. The top-view SEM images of D2 tool steel substrates, after ion etching in the BAI, CC7 and AIP deposition systems, show that the etching efficiency of individual methods is different (Figure 2). The intensity of ion etching is reflected in the height of the step at the sites of protruding $Cr_7C_3$ carbides, which have a much smaller etching rate than the ferrous matrix. Outside the carbide area, the surface of substrate etched in the BAI system looks rather smooth, while it is rougher for samples etched in the CC7 and AIP systems. All topographic irregularities formed during ion etching are transferred to the coating surface and even magnified due to the geometrical shadowing effect. The protruding carbides of the substrate are, therefore, also visible in the SEM images of the TiN coating surfaces (Figure 3).

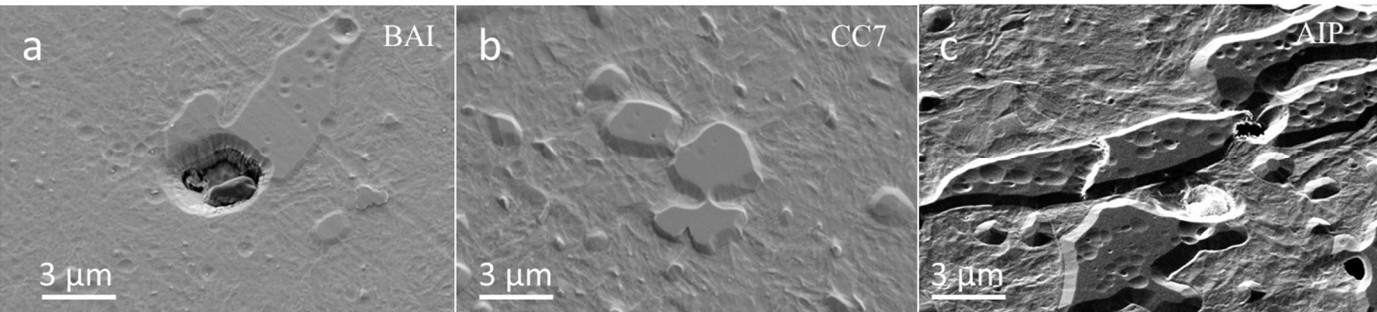

**Figure 2.** Top-view SEM images of D2 tool steel substrates after ion etching in BAI (**a**), CC7 (**b**) and AIP (**c**) deposition systems.

The topography of the coating surface is most affected by the growth defects (e.g., nodular defects, craters, droplets) formed during the deposition process [29]. The growth defects occur at sites of substrate topographical irregularities and at foreign particles that remain on the substrate surface after substrate pretreatment (cleaning, ion etching), and a part of them originates from the deposition process. The distribution of growth defects (nodular defects, droplets, pinholes, craters) on the coating surface and their shape and size were analyzed using a SEM microscope (Figure 3) and 3D stylus profilometer (Figure 4). The top-view SEM images of the BAI, CC7 and AIP TiN coating surfaces were taken at low (Figure 3a–c) and high magnifications (Figure 3d–f). In SEM images recorded at low magnification, we can see that growth defects are unevenly distributed over the coating surface. The lowest concentration of growth defects was observed on the BAI coating, while the highest one was on the AIP coating (Figures 3a–c and 4). High magnification SEM images (see insets in Figure 3d–f) of the area without growth defects reveals a dimpled surface topography for BAI coating, while the surfaces of the CC7 and AIP coatings are a little bit rougher, similar to substrate surfaces after ion etching.

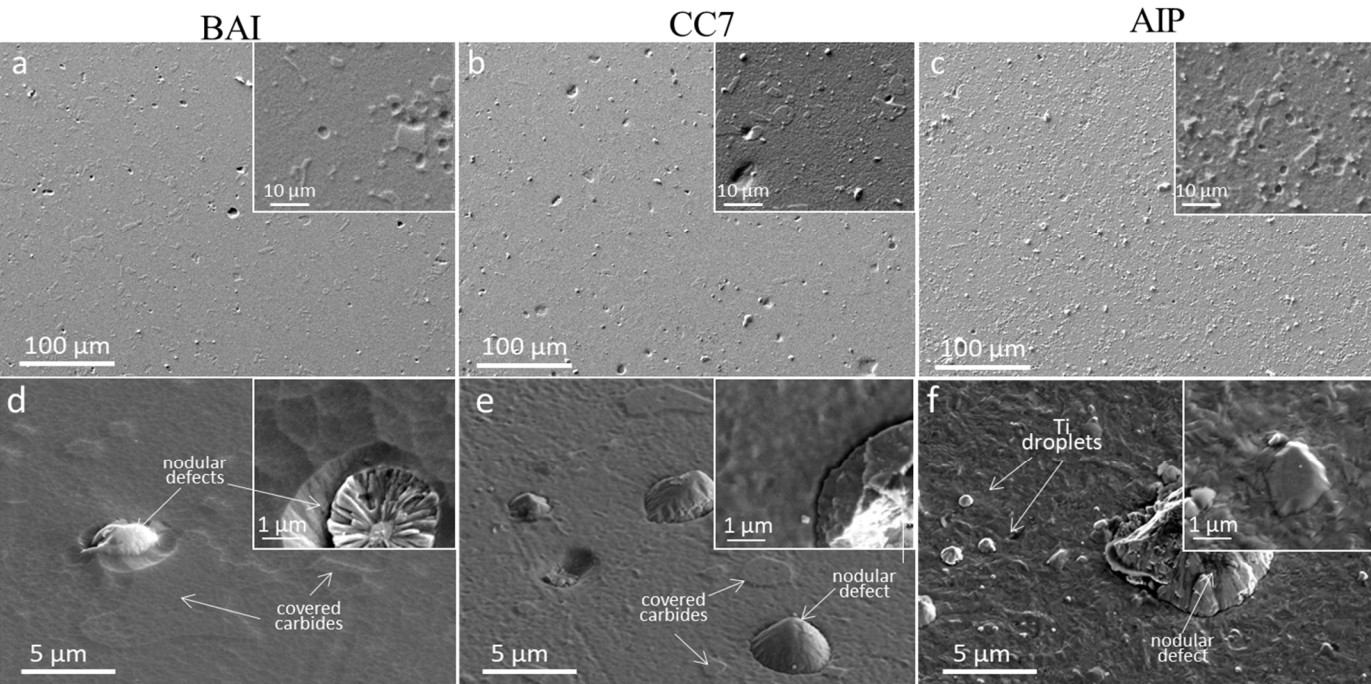

**Figure 3.** Top-view SEM images of TiN hard coatings deposited onto D2 tool steel substrate in BAI, CC7 and AIP deposition systems. The upper SEM images (**a**–**c**), including insets, were recorded at low magnifications, while the lower images (**d**–**f**), including insets, were taken at high magnification and by tilting the sample approximately 20°.

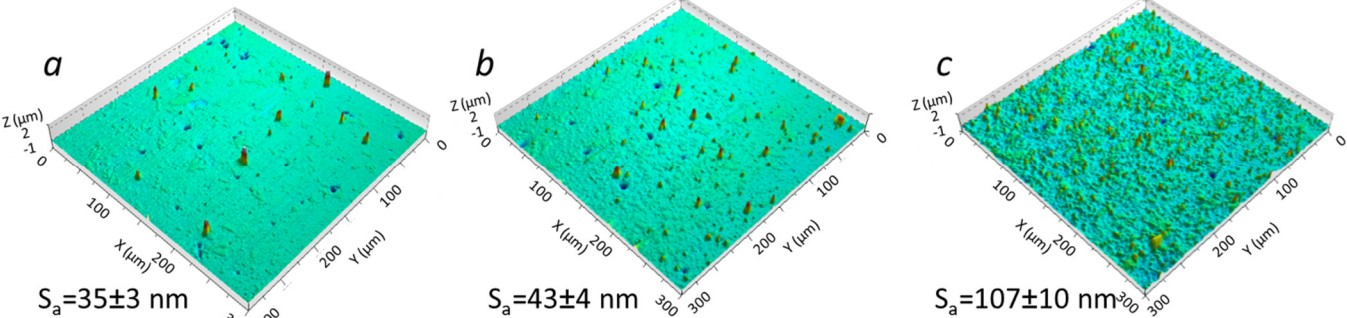

**Figure 4.** Three-dimensional profile images of TiN coatings deposited onto D2 tool steel substrate in BAI (**a**), CC7 (**b**) and AIP (**c**) deposition systems. The surface roughness values $S_a$ are also added [26]. The nodular defects are the sharp peaks, while the blue dots are craters. The scan area was 300 μm × 300 μm, while the z-scale was 3 μm.

A 3D stylus profilometer was used to quantify the density of growth defects. Due to the relatively low surface density of growth defects and their uneven distribution, only large scanning areas provide a reliable result. In our case, the evaluation area was 1 mm$^2$ with a resolution of 2 μm in the x-direction and 0.2 μm in the y-direction, while the effective vertical resolution was around 5 nm. In order to distinguish the growth defects from the background, a threshold value has to be determined, and it should be larger than the step height of protrusions at carbide inclusions (which is typically about 200 nm). In our experience, a reasonable threshold value is 0.5 μm. The sharp peaks in Figure 4 are in most cases the nodular defects. The surface density of nodular defects, evaluated using these conditions, was about 180, 360 and 610 defects/mm$^2$ for the BAI, CC7 and AIP TiN coatings, respectively. On the AIP coatings, the growth defects originated mainly from the metal droplets, formed during the coating deposition process. Droplet-like defects (Figure 3f)

were much smaller than the other protrusions (Figure 3d,e), but their density was much higher. Therefore, the roughness of the AIP coating was much higher than that of the other two. Parts of the droplets were torn out of the coating due to the high compressive stresses, leaving small craters (see inset in Figure 3c).

In general, the surface topography of a coating can be described with different surface roughness parameters [30]. The most commonly used surface roughness parameter is $S_a$, which is the absolute value of measured height deviations from the surface mean area. However, it does not give any information on the wavelength and is not sensitive to small changes in profile. Contrarily, the *surface skewness* $S_{sk}$ is a measure of the asymmetry of the surface profile from the surface mean line. Zero skewness reflects symmetrical height distribution, while positive and negative values of $S_{sk}$ indicate a prevalence of peaks and valleys, respectively. Due to carbide protrusions formed during polishing of bare D2 substrates, the value of the parameter $S_{sk}$ was positive (Figure 5). It remained positive also after ion etching and deposition in all three deposition systems. However, after ion etching in BAI, CC7 and AIP systems, the parameter $S_{sk}$ decreased from 2.86 for the polished substrate (where protruding carbides predominate) to 1.8, 1.3 and 0.53, respectively. This means that due to the inhomogeneity of the D2 tool material and the consequently different etching rates of various phases and grains with different orientations, both shallow depressions and protrusions were formed. This is reflected in both higher roughness $S_a$ and lower skewness $S_{sk}$. After deposition, the skewness parameter significantly increased due to the formation of growth defects—the most for the BAI coating and the least for the AIP coating. This means that nodular defects predominated in the BAI coating, while there were also many craters in the CC7 and especially in the AIP coatings. Since there were many metal droplets on the surface of the AIP coating, the corresponding skewness was smaller for this coating. Useful information also can be obtained from the roughness parameter $S_{ku}$ (*kurtosis*), which highlights whether the surface profile has steep or rounded peaks and valleys. For relatively flat surfaces, the $S_{ku}$ value is smaller than 3; it is equal to 3 for perfectly Gaussian surfaces, while it is larger than 3 for surfaces with sharp peaks and valleys. In our case, the kurtosis was more than 3 for polished substrate surface, as well as for all substrates after ion etching and deposition in the BAI, CC7 and AIP systems. Kurtosis decreased after ion etching (the most in the CC7 system and the least in the BAI) but increased significantly after deposition (the most for BAI coating and the least for the AIP one). This means that the peaks formed during ion etching at sites of carbide and nonmetallic inclusions were less sharp, while during the deposition process, growth defects were formed, significantly increasing the $S_{ku}$ roughness parameter.

*2.3. X-ray Diffraction Analysis*

Characterization of the crystallographic texture, grain size and crystal phases in the coatings was performed using XRD diffraction analysis (Figure 6). In addition to peaks originating from the TiN coating, substrate peaks (labeled »s« in Figure 6) were also present. The analysis showed that all three TiN coatings had a B1 NaCl crystal structure with varying degrees of preferred orientation. The positions of TiN diffraction peaks could be indexed to the face-centered cubic phase structure of TiN (JCPDS file No. 38-1420) with lattice constant a = 0.4241 nm [31]. Measured values of interplanar spacing $d_{hkl}$ and the corresponding lattice parameters are given in Table 2. The spacings $d_{hkl}$ for all reflecting planes (except (200)) parallel to the surface were 0.2–0.9% larger than the reference values obtained from a randomly oriented strain-free standard sample. This indicates the coatings were in a state of inhomogeneous compression stresses. The lattice parameters of the TiN coatings were in the range from 0.4219 to 0.4267 nm. Therefore, they slightly deviated from the bulk value of 0.424 nm. In the case of stoichiometric TiN coatings, this deviation can be explained by intrinsic stresses that can be caused by small grain size, high defect concentration, and/or interstitial incorporation of argon or nitrogen.

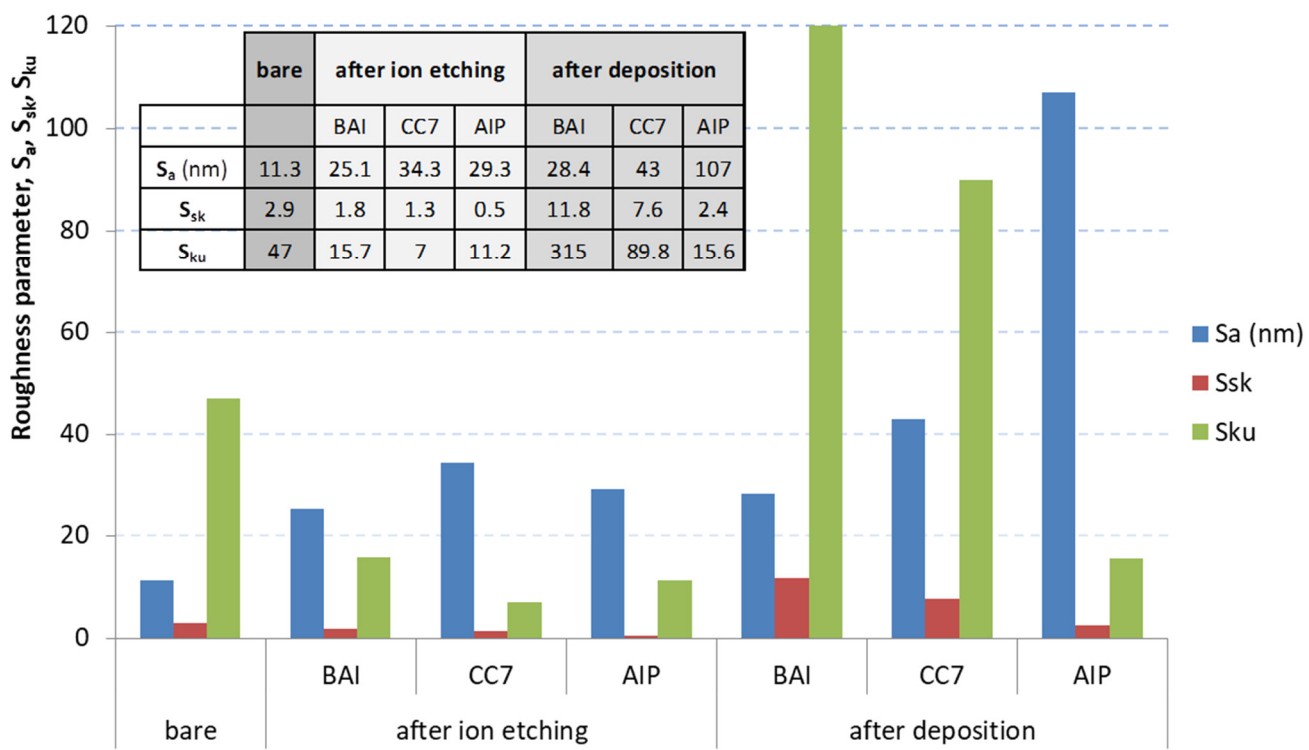

**Figure 5.** Roughness parameters ($S_a$, $S_{sk}$ and $S_{ku}$) of D2 substrate, after polishing, after ion etching, and after deposition of TiN coatings in BAI, CC7 and AIP systems.

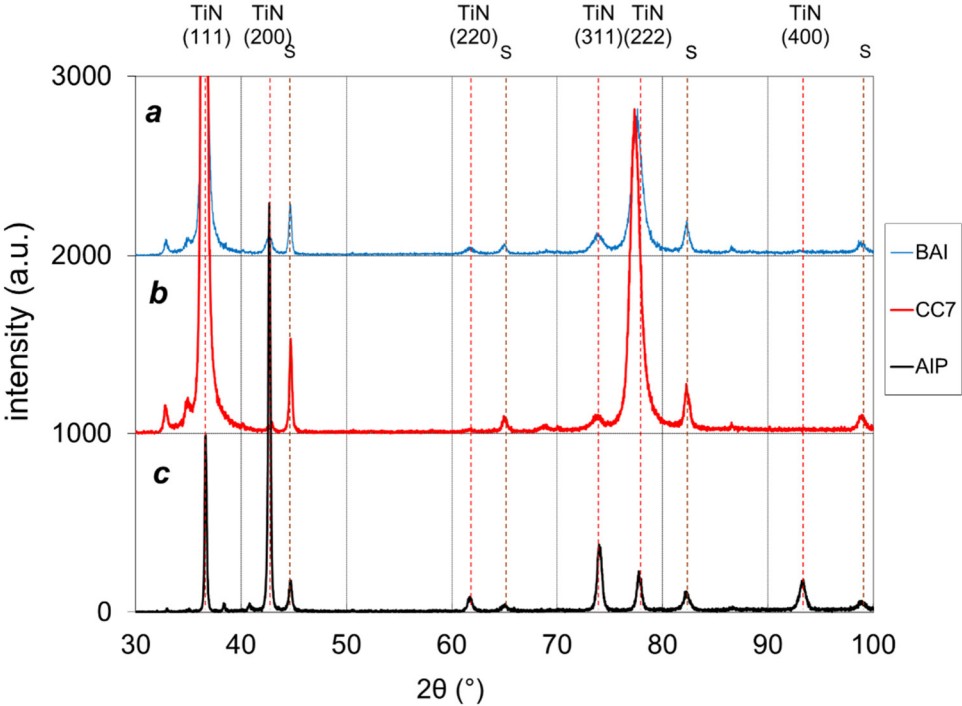

**Figure 6.** XRD spectra of TiN coatings deposited by low-voltage electron beam evaporation (**a**), magnetron sputtering (**b**), and cathodic arc deposition (**c**).

**Table 2.** Results of X-ray diffraction analysis of TiN coatings deposited by three different deposition methods: low-voltage electron beam evaporation (BAI), unbalanced magnetron sputtering (CC7) and cathodic arc deposition (AIP).

| - | Interplanar Distance, $d_{hkl}$ (nm) Lattice Parameter, $a_{hkl}$ (nm) | | | | FWHM (°) | | | Relative Intensity (%) | | |
|---|---|---|---|---|---|---|---|---|---|---|
| Deposition process | $d_{111}$ $a_{111}$ | $d_{200}$ $a_{200}$ | $d_{220}$ $a_{220}$ | $d_{311}$ $a_{311}$ | $\beta_{111}$ | $\beta_{200}$ | $\beta_{220}$ | $I_{111}/\Sigma I_{hkl}$ | $I_{200}/\Sigma I_{hkl}$ | $I_{220}/\Sigma I_{hkl}$ |
| Reference (6-0642) [31] | 0.2449 | 0.212 | 0.1496 | 0.1277 | - | - | - | 26.9 | 35.8 | 19.7 |
| BAI | 0.2459 0.4259 | 0.2118 0.4236 | 0.1502 0.4248 | 0.1281 0.425 | 0.5 | 0.71 | 0.95 | 75 | 1.8 | 0.6 |
| CC7 | 0.2462 0.4265 | 0.2110 0.4219 | 0.1502 0.4249 | 0.1282 0.4252 | 0.45 | 0.55 | 0.72 | 83 | 0.2 | 0.1 |
| AIP | 0.2451 0.4245 | 0.2117 0.4234 | 0.1502 0.4248 | 0.128 0.4244 | 0.23 | 0.27 | 0.51 | 24.8 | 55.6 | 1.8 |

The full width at half-maximum (FWHM) peak broadenings $\beta_{hkl}$ are also given in Table 2. The measured values showed that peak broadening was the highest for BAI samples ($\beta_{111}$ = 0.5°) and the lowest for AIP samples ($\beta_{111}$ = 0.23°). A broader peak could be explained by smaller grain size and/or the presence of inhomogeneous microstrains in the crystallites. The high ratio of $\beta_{222}/\beta_{111}$ indicates that the main contribution to peak broadening can be attributed to the inhomogeneous strain rather than small grain sizes. It is commonly known that the ion bombardment of growing coatings induces intrinsic compressive stresses through the entrapment of nitrogen and/or argon ions in non-equilibrium on interstitial (tetrahedral) lattice sites. The phenomenon is favored at low deposition rates and low deposition temperature.

The intensities of TiN reflections are also very different from those of a randomly oriented sample. There are, however, clear differences among the samples in the degree of preferred orientation. The preferred orientation of the coatings could be quantified by evaluating integrated intensities of four apparent peaks, namely, (111), (200), (220), and (311). Texture coefficients for all three types of coatings are included in Table 2. The degree of preferred orientation of the coatings is given by the texture coefficient $\gamma_{hkl}$ ($\gamma_{hkl}$ = $I_{hkl}/\Sigma I_{hkl}$, where $I_{hkl}$ is the intensity of a specific peak and $\Sigma I_{hkl}$ is the sum of the intensities of all detected TiN reflections). The BAI and CC7 samples had 75% and 83% degrees of (111) preferred orientation parallel to the substrate surface, respectively. All other diffraction reflections contributed very little to the total diffracted intensity. The (111) orientation became a little bit less dominant in the BAI sample. AIP samples, in contrast, had (200) preferred orientation ($\gamma_{200}$ = 55.6%) with a significant (111) component (25%) and much smaller (220) and (311) peaks (1.8% and 9%, respectively).

The preferred orientation of TiN coatings is determined by the minimization of the overall free energy per surface coating area, which resulted from the competition between the surface energy, the strain energy, and the stopping energy of different lattice planes [32,33]. For TiN coatings with a NaCl crystal structure, the (200) plane has the lowest surface energy, the (111) plane has the lowest strain energy, and the stopping energy is minimal in the (220) plane. The intensive ion bombardment of the growing coating increases the accumulation of strain energy in coatings and the (111) plane becomes the preferred orientation [34]. It should be stressed that the surface energy is independent of film thickness, whereas the strain energy increases linearly with the thickness of the film [35]. Therefore, at lower thickness, the surface energy term is significant and (100) orientation with minimum surface energy may be expected. At a larger film thickness, the strain energy becomes dominant and (111) preferred orientation is the result of relieving the strain energy. The stopping energy (defined as energy of ions deposited due to channeling effect) is minimal in the (220) plane, while <001> direction is the most open channeling

direction, where the energy of impinging ions is distributed over larger volumes. Channeling energy is dominant (220 preferred orientation) only when the deposited ion energy is sufficiently high (bias voltage greater than −100 V). The texture of TiN coating can be changed either by increasing the ion energy (by increasing bias voltage) or by increasing the surface mobility of deposited species (by increasing the substrate temperature or ion-to-atom flux ratio) [36,37]. Increasing the adatom mobility through bias voltage may enable atoms to rearrange and thus form other textures to reduce competing energies. For example, Greene et al. found that a moderate ion bombardment of TiN films, prepared by conventional magnetron sputtering technique, leads to the (111) texture, while an intense ion bombardment leads to (001) texture [38].

X-ray structural analysis of TiN coatings showed that the dominant crystal plane parallel to the plane of the substrate was (111) for the BAI and CC7 samples and (200) for the AIP sample. These differences were most likely related to ion irradiation conditions during all three deposition processes [39]. Namely, the comparison of the three deposition techniques used for the preparation of the TiN coatings shows that the ion current densities on the substrates were about 2, 2.5 and 3 mA/cm$^2$ for CC7, AIP and BAI samples, respectively. On the other hand, the average deposition rate was 0.85, 0.36 and 1.56 nm/s for the CC7, BAI and AIP coatings, respectively. The bias voltage was the highest for the BAI sample (−125 V), while it was −100 and −90 V for the CC7 and AIP sample, respectively. Both ion energies and ion-to-atom flux ratio ($j_i/j_{Ti}$) were the highest in the case of the BAI deposition technique. That could be the reason why the BAI TiN coating had the most pronounced (111) preferential orientation.

The preferred crystallographic orientation has a strong influence on the resulting physical properties. Coating texture affects the mechanical behavior of the coatings. Thus, it has been reported that TiN coating with (111) preferred orientation possesses the highest hardness and superior resistance to abrasion and wear [40,41]. Ponnon et al., found that the (200) preferential orientation is more conductive than the (111) one [42]. They also found that the sample with a predominant (200) orientation had a considerably smoother surface than the one with (111) orientation.

### 2.4. Mechanical Properties of TiN Coatings

The tribological performance of hard coatings is strongly related to their hardness (H) and elastic properties (Young's modulus, E). In order to perform well in a tribological application, a coating should be both hard and ductile. Thus, high hardness and low modulus are desirable to enhance fracture toughness and elastic strain to failure (high yield strength). A high H/E ratio (*elasticity index*) is often a reliable indicator of good tribological behaviors of coatings [43,44]. The H/E ratio defines the amount of energy that a coating material can absorb without permanent damage. In general, it can be expected that the friction coefficient and wear rate of the coatings are improved as their Young's modulus is reduced. Another important parameter in the investigation of the relation between the mechanical and tribological behavior of coatings is the ratio $H^3/E^2$ (*plasticity index*). Namely, the coating resistance to plastic deformation is proportional to the ratio $H^3/E^2$. This means that the plastic deformation in coating materials with high hardness H and low modulus E is reduced because low modulus E allows the given load to be distributed over a wider area of contact.

Both the hardness and elastic modulus of TiN coatings prepared by BAI, CC7 and AIP deposition techniques were determined from nanoindentation measurements. The indents were all performed on flat areas free from visible growth defects. The measured values of hardness and elastic modulus, as well as the calculated H/E and $H^3/E^2$ ratios, are given in Table 3. It can be seen that the measured hardness was considerably lower for the AIP coating compared to the BAI and CC7 coatings. The main factor leading to the high hardness of both the BAI and CC7 TiN coatings was their fine-grained, compact and dense microstructure, while the AIP coating was characterized by a pronounced columnar structure. An additional reason for higher microhardness could be the (111) preferential

growth of the BAI and CC7 coatings. On the other hand, fine-grained microstructure does not significantly affect elastic modulus. We found that all three coatings were characterized by similar elastic modulus values. Consequently, the elasticity index H/E and plasticity index $H^3/E^2$ for the AIP coating were lower in comparison with those of the BAI and CC7 TiN coatings. On the other hand, this meant less wear resistance of the AIP coating, which confirmed measurements of the wear coefficient (see Section 2.5). It can be expected that both the elasticity and plasticity index of coatings depend on the crystal structure, texture, internal stresses and microstructure. However, the wear resistance of a coating depends not only on the index of elasticity and plasticity, but, as we will see later, also on other factors (e.g., surface topography, tribo-oxidation processes).

**Table 3.** Hardness (HV0.05), elastic modulus (E), H/E ratio and $H^3/E^2$ ratio of TiN coatings deposited in BAI, CC7 and AIP deposition systems.

| - | H (GPa) | E (GPa) | H/E | $H^3/E^2$ (GPa) |
|---|---|---|---|---|
| **BAI** | $29 \pm 2$ | $376 \pm 16$ | $0.080 \pm 0.008$ | $0.17 \pm 0.05$ |
| **CC7** | $28 \pm 2$ | $377 \pm 16$ | $0.080 \pm 0.008$ | $0.15 \pm 0.05$ |
| **AIP** | $25 \pm 2$ | $378 \pm 16$ | $0.066 \pm 0.007$ | $0.11 \pm 0.035$ |

*2.5. The Tribological Behavior of TiN Coatings*

Although the surface topography of coatings and surrounding atmosphere strongly affect the tribological performance of PVD TiN coatings, literature that systematically addresses these subjects is scarce [23–25,45–48]. Therefore, in this study, we performed more systematic tribological tests of BAI, CC7 and AIP titanium nitride coatings in three different atmospheres. In particular, the dependence of the friction coefficient on the type of atmosphere in the initial phase of the sliding test (running-in period) was analyzed in more detail.

2.5.1. The Influence of Coating Topography on Friction and Wear

One of those properties of TiN coatings that has the greatest impact on its tribological behavior is surface topography. As discussed above (Section 2.2), the topographies of BAI, CC7 and AIP TiN coatings are rather different. Therefore, we would expect that these differences are reflected in their tribological behavior as well. In order to study the influence of topography during the tribological test, the formation of an oxide layer must be prevented. This can be achieved by performing the sliding test in a nitrogen atmosphere. We assumed that in the absence of such a layer, the tribological response of the TiN coating would be due mainly to its surface topography. Figure 7 shows representative friction curves obtained during the sliding test of an alumina ball on as-deposited and polished BAI, CC7 and AIP TiN coatings in the nitrogen atmosphere. We can see that different coatings showed different levels of friction development in the initial phase of the sliding test, as well as different levels of friction in the steady-state period. In the initial phase of the sliding test, the friction coefficient rapidly increased and reached a value slightly below 0.3 for the BAI coating after 15 cycles, slightly above 0.3 for the CC7 coating after 20 cycles, and 0.5 for the AIP coating after 50 cycles. Immediately afterwards, the coefficient of friction was reduced to 0.2 for the BAI and CC7 coatings, and to about 0.45 for the AIP coating. Afterwards, the coefficient of friction gradually increased with a small fluctuation to a steady value, which was 0.47, 0.44 and 0.57 for the BAI, CC7 and AIP coatings, respectively. A similar dependence of the coefficient of friction in the initial phase of the sliding test was reported by other authors, but for different types of hard coatings [47,49]. The initial rapid increase and then immediate decrease in the friction coefficients for the as-deposited coatings were due to decreased ploughing action of protrusions on the coating surfaces. This explanation was confirmed by measurements of friction coefficients on the post-polished coatings. Namely, on friction curves of the post-polished coatings (Figure 7), no initial rapid change could be observed. This was because polishing reduced both the height of protrusions and

also their number. The initial value of the coefficient of friction was 0.2 for all three types of TiN post-polished coatings. This was also the value achieved on as-deposited BAI and CC7 coatings after the initial few tens of sliding cycles, while this was not the case for the AIP coating, which we will try to explain later. The running-in period of post-polishing BAI and CC7 coatings was significantly shorter as compared with the as-deposited ones, while there was no difference for AIP coating. The wear rate of the post-polished coating was much lower than that of the as-deposited one. In some cases, it decreased even more than 50%, depending on the degree of polishing. Namely, post-polishing of the as-deposited TiN coating significantly reduces the material pick-up tendency due to the generation of smoother, less abrasive, surfaces [46]. We have to mention another phenomenon that is due to post-polishing of coatings and that can affect their tribological behavior. Namely, due to high shearing stresses during post-polishing of a coating, a portion of the nodular defects and droplets are torn out, and craters are formed on the surface. During the sliding process, some newly formed wear particles are trapped in such craters (particle hiding); therefore, abrasive wear is reduced.

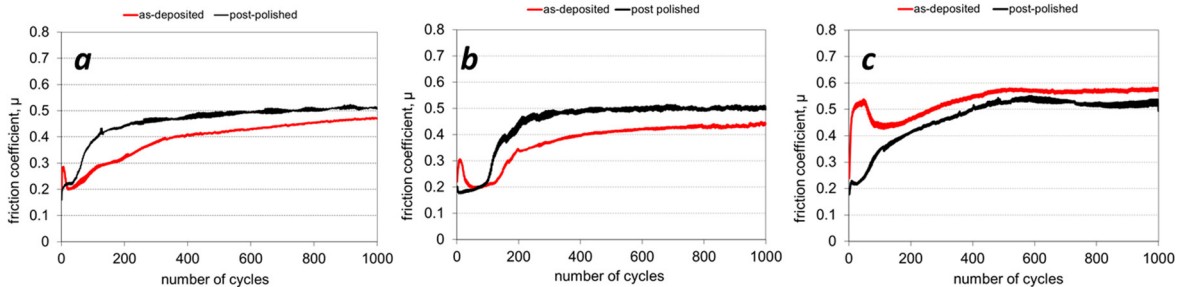

**Figure 7.** Friction curves of as-deposited and polished BAI (**a**), CC7 (**b**) and AIP (**c**) samples (TiN/D2).

In the initial phase of the sliding test, two mechanisms affected the friction and wear. We had to consider that the initial contact spots were at the coating protrusions (e.g., nodular defects), which cause friction due to ploughing action on the softer counter-body surface (alumina ball). A slight increase in the friction after the first sliding cycle was due to the increase in alumina ball contact area. The first contact between the flat coating surface and the alumina ball, which moved relative to each other, always took place at the highest peaks (nodular defects) of the surfaces, while the contact conditions were very complex. Protruding nodular defects cause abrasive wear of the counter material. Due to the ploughing and material pick-up effects, the friction coefficient increases. The pressure at these contact spots is very high due to the small actual contact area (about 10% of the surface area) [20]. However, the real area of contact depends not only on surface roughness but also on the applied normal load. Thus, for example, the contact area increases with the normal load if the contact is elastic. High pressure and shear stresses cause the collapse of the most protruding nodular defects into small fragments that are released into the wear track [46,50,51]. Therefore, the nodular defects represent the primary source of hard abrasive particles in the sliding contact during the running-in period. These particles have a strong effect on friction and wear because along with protrusions, they abrade the softer surface of the counter-body material.

The second reason for friction increase is the interlocking of asperities, on both the coating and counter-body surfaces. When two surfaces slide relative to each other, the friction force is not constant, because the motion changes periodically between the adhesion and sliding (i.e., the *stick-slip phenomenon*). After removal of the highest nodular defects at the beginning of sliding, contact is formed at the lower protrusions (e.g., sites of carbides and nonmetal inclusions) [51] (Figure 8). Although all of these asperities are much smaller, they cover a much larger surface area. They also offer additional anchoring points for counter-body material transfer. We found that a considerable amount of debris had already collected in front of such protrusions after a few cycles.

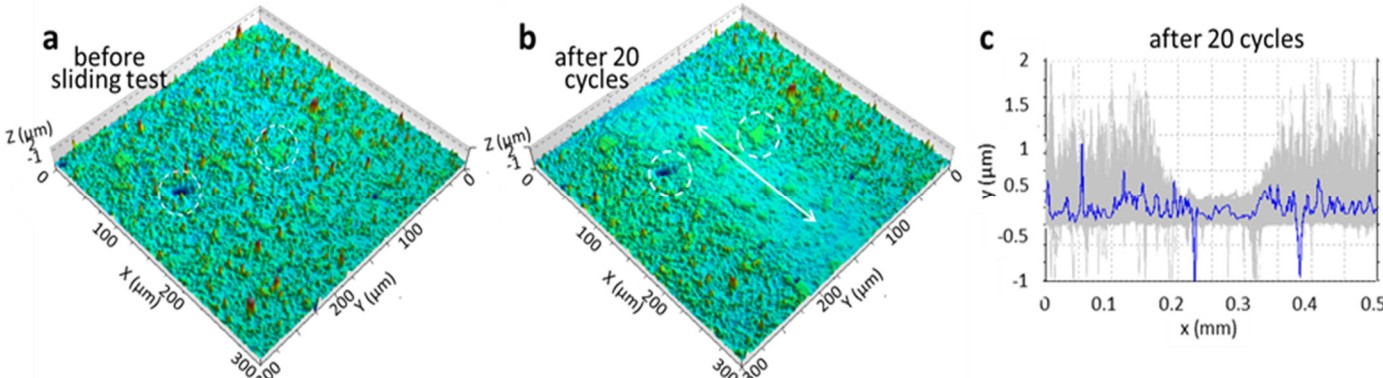

**Figure 8.** Three-dimensional profile images of the wear tracks on AIP TiN hard coating on D2 tool steel substrate before (**a**) and after 20 cycles of sliding against an alumina ball (**b**). The scanning area was 0.3 mm × 0.3 mm. Both 3D-profile images were taken from the same area on the substrate surface (the same sites before and after the test are labeled with the white circles). The sliding direction is indicated by arrows. The protrusions at sites of carbides are still visible in the wear track. The sharp peaks are the nodular defects, while the blue dots are craters (note the strong exaggeration in z-scale). (**c**) A series of a numerous line profiles were recorded perpendicular to the wear track and stacked together. The blue profile represents the level of coating, while grey peaks belong to protrusions.

During further sliding, all protrusions are gradually removed and a smooth surface is formed. As long as protrusions are present on the coating surface, the friction coefficient does not reach the steady-state. From the explanation described above, we can conclude that the rougher the coating is, the longer the time that is required to obtain the steady-state value of the friction. The period until a conformal sliding contact is formed is called the *running-in phase*. In this period we can expect a transition from *mechanical wear-dominated friction* to *adhesion-dominated friction*. The larger contact surface is reflected in a higher friction coefficient due to the higher adhesion component of the friction. It is also valid that as the contact area increases, the contact pressure decreases.

As already mentioned above, the initial value of the coefficient of friction of the as-deposited AIP coating was higher compared to that of the BAI and CC7 coatings. The reason for the different tribological behavior of the AIP coating in comparison to the BAI and CC7 coatings could be related to titanium droplets on the surface of the AIP coating. It can be expected that these relatively soft droplets can increase the adhesion component of friction. Because they are incorporated throughout the entire thickness of the coating, droplets also affect the friction coefficient after the top layer has been removed during the sliding test. This could be the reason why the steady-state value of the friction coefficient was highest for the as-deposited AIP coating (around 0.58), while it was around 0.45 and 0.47 for the CC7 and BAI coatings, respectively. The steady-state value was reached the fastest with the CC7 as-deposited coating (after about 300 cycles) and the slowest with the BAI coating (after about 500 cycles).

Still another aspect of the sliding contact should be mentioned. The characterization of microstructure (Section 2.1) showed that all three coatings had a more or less pronounced columnar microstructure. The columns grew normal to the coating substrate interface, while the individual columns were not in close contact. We could expect that the material transferred from the counter-body ball would be trapped in the open regions between the columns. In general, each column was characterized by a domed end-cap that also may have abraded the ball material as it cycled over the coating surface, resulting in the transfer of the plastically deformed material to it. Therefore, the more columnar and porous the coating is, the greater the contribution of the adhesive friction component that can be expected. The microstructures of our BAI, CC7 and AIP TiN coatings appeared quite compact; therefore, we do not believe that this mechanism played a significant role during the tribological test.

### 2.5.2. Friction and Wear in Different Surrounding Atmospheres

The tribological processes change drastically if the sliding test takes place in ambient air or in an oxygen atmosphere (Figures 9–11). In both cases, the tribological process was not affected only by the material transfer but also by the formation of the tribo-oxide layer. Oxides form in the sliding contact very quickly due to heating induced by friction. However, high compressive stresses appear in the oxide layer due to the large difference in the oxide layer molar volume in comparison to the molar volume of the nitride layer. These stresses, together with high shear forces, cause the formation of cracks in the oxide film and finally its spallation. Only small patches of oxides in the wear track can be observed. Delaminated oxide fragments can act as abrasive particles, which cause a significant increase in the coefficient of friction and wear rate. Repeated sliding results in the accumulation of oxide particles in the wear track (in the form of *roll-like debris*). Tribological tests performed in different atmospheres (Figures 9 and 11a) confirmed that all types of TiN coatings have the highest coefficient of friction in an oxygen atmosphere (about 1), while it is much smaller if the sliding test is performed in nitrogen (about 0.55) or in ambient air (about 0.4). Not only the friction coefficient but also the wear rate strongly depends on the type of atmosphere in which the test takes place (Figures 10 and 11). The wear coefficient is the highest in an oxygen atmosphere, lower in ambient air, and the lowest in nitrogen. The use of a nitrogen atmosphere suppresses the formation of an oxide layer, resulting in a strong reduction in the TiN coating wear rate. We believe that, besides the cooling effect, this mechanism is active in tool protection during cryogenic machining using liquid nitrogen.

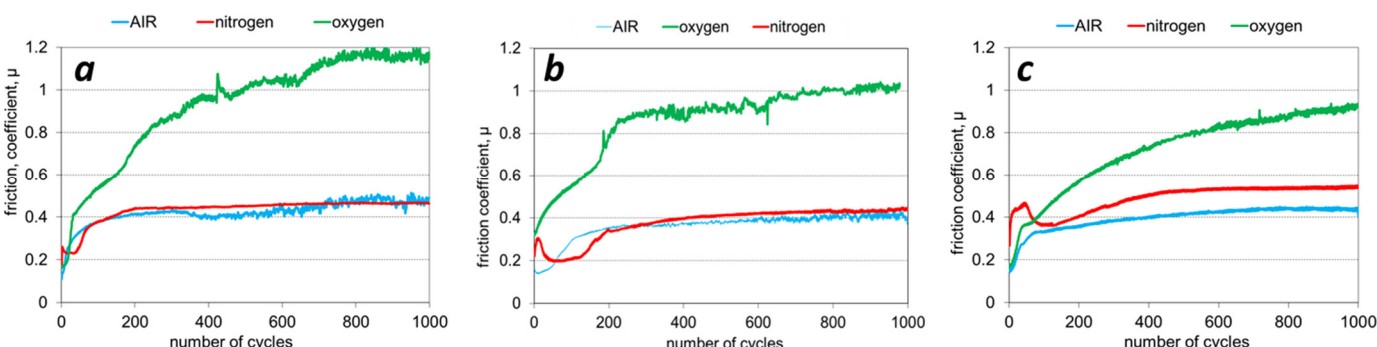

**Figure 9.** The friction curves of as-deposited BAI (**a**), CC7 (**b**), and AIP TiN coatings (**c**) for the tests performed at room temperature in different atmospheres (ambient air, nitrogen and oxygen) under a load of 5 N.

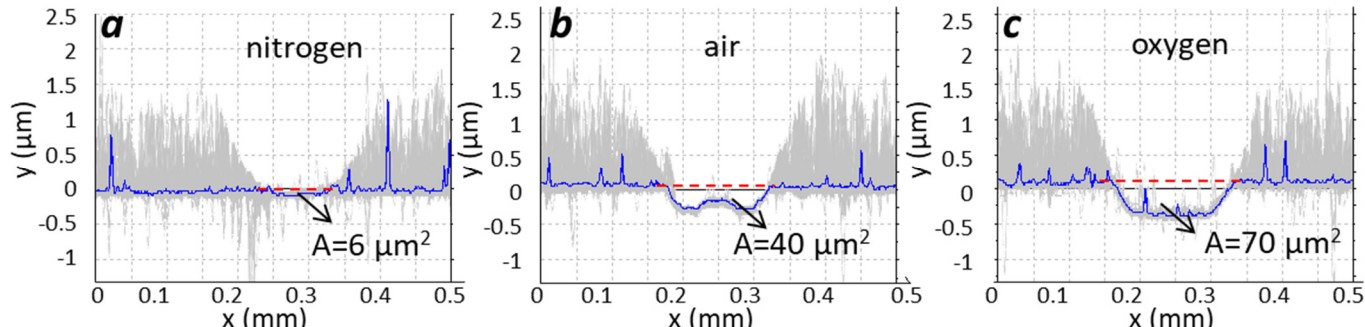

**Figure 10.** Series of 250 line profiles stacked together from the recorded profiles. The wear track of BAI TiN coating is shown for different surrounding atmospheres ((**a**)—nitrogen, (**b**)—air, (**c**)—oxygen). The scanning area was 0.5 mm × 0.5 mm. The blue profile represents the level of coating, while grey peaks represent protrusions. During the sliding test (1000 cycles), the protrusions were removed and the wear tracks with various depths were formed. Note the strong exaggeration in z-scale. "A" is the cross-section area of the wear track.

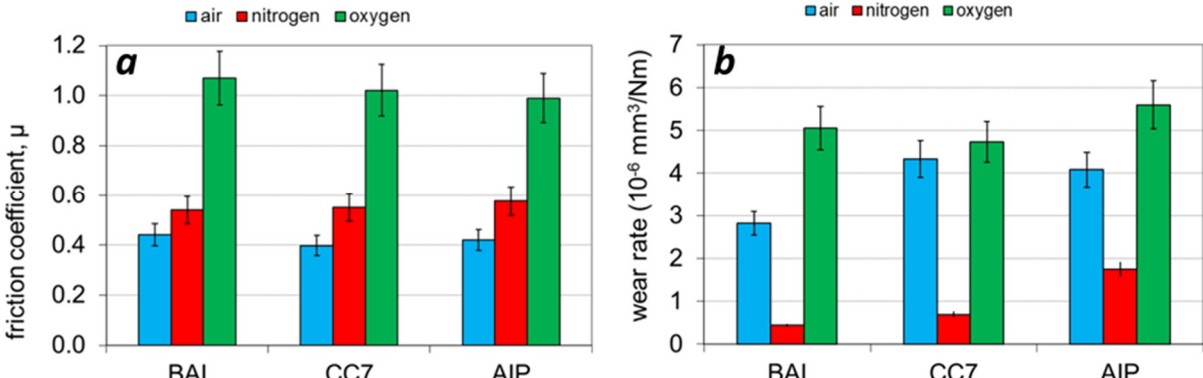

**Figure 11.** The friction coefficients (**a**) and the wear rates (**b**) in the ambient, oxygen and nitrogen atmospheres for three different coatings.

The influence of the surrounding atmosphere on the formation of the oxide layer was also analyzed by the following experiment. The sliding tests performed on the BAI TiN coating in the nitrogen, oxygen and ambient air atmospheres were interrupted after 50 cycles for 30 min (Figure 12a). After 30 min, the sliding tests were continued, and a different tribological response of TiN coating in the studied atmospheres was observed. It could be observed that such an interruption did not significantly affect the tribological process when the test took place in a nitrogen atmosphere. However, this was not the case in the tests performed in oxygen or ambient air atmosphere. In these cases, the coefficient of friction decreased to the initial value after each interruption and then, after a short running-in period, increased again to the value before the interruption. This indicates that a thin native oxide layer formed on the wear track during the interruption period.

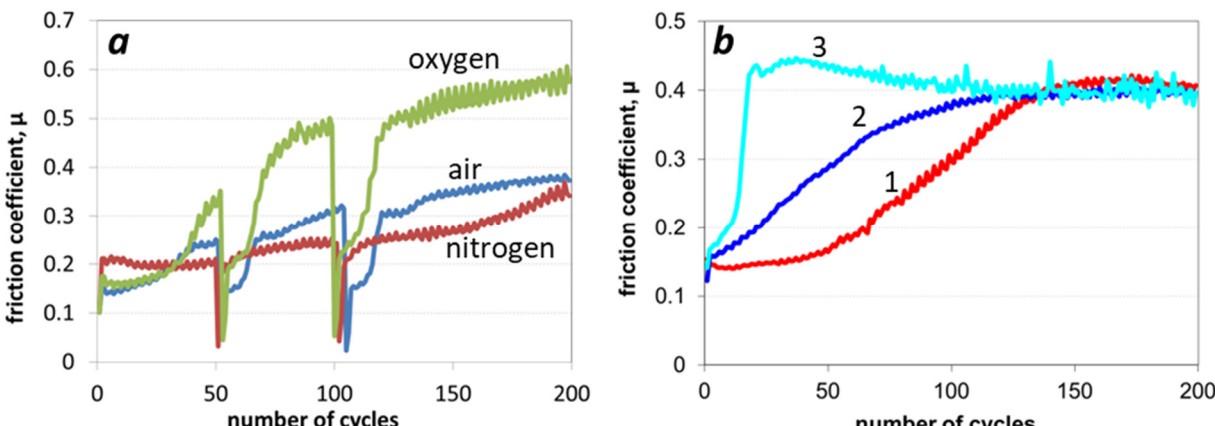

**Figure 12.** (**a**) The sliding test on sputter-deposited TiN coating was interrupted after 50 laps for 30 min and then continued using same conditions. After sliding test was continued, different friction development was observed in the three studied atmospheres. (**b**) The running-in period depends on the state of coating and ball surfaces: the sliding test was performed in ambient air on the fresh coating surface and a new spot on the alumina ball (1); on the worn ball surface and a new track on the coating surface (2); on a new spot on the alumina ball and the track formed on the coating surface during the previous test (3).

In the next experiment, we tried to separate the influence of the coating surface and alumina ball surface on the friction during the running-in period of the sliding test. The following three sliding tests in the ambient air atmosphere were performed: (i) in the first test, we selected a fresh as-deposited coating surface and a new spot on the alumina ball; (ii) in the second test we used the worn ball surface, but a new track on the coating surface; (iii) the third test was performed in a track formed during the previous test, while

a new fresh surface on the alumina ball was selected. The results of friction coefficient measurements are shown in Figure 12b. In the case of a fresh coating surface and a new spot on an alumina ball, the duration of the running-in period was approximately 150 cycles. If we restarted the sliding test in the track formed during the previous test with fresh ball surface, then the running-in period was shortened to 20 cycles. In the case of worn alumina surface and fresh coating surface, the duration of the running-in period was about 115 cycles. Based on these results, we can conclude that the duration of the running-in period depends mainly on the coating surface conditions.

Wear rates given in Figure 11 are average values of measurements from a large number of samples. It should be noted, however, that all of these coatings differed in surface roughness. We have to consider that the surface roughness of the TiN coating does not depend only on the deposition method, but largely depends on the surface density of growth defects. The formation of such defects is a spatially localized and sporadic process [52]. Therefore, significant differences in surface roughness were observed not only for samples within the same batch, but even between samples from different batches. Figure 13 shows, for example, the peak (nodular defect) density distribution for BAI TiN hard coating. From a hundred measurements, we performed statistical analysis. The samples were divided into 14 classes with a width of 50 peaks/mm$^2$ ranging from 0 to 700 peaks/mm$^2$. For any series of samples, we constructed 14 classes (x, y), where x was the median number of defects of the class, and y was the number of samples falling into this class. In this way, we obtained the peak density distribution, which presents the samples per peak class versus the number of peaks. The data on the $y$-axis were normalized. For fitting of measurements, we used the Poisson distribution. In order to study the influence of surface roughness on the tribological response of TiN coatings prepared by one of three deposition methods, we selected the samples with different surface densities of growth defects. The surface density distribution of nodular defects for BAI TiN coatings showed a pronounced peak at 200 defects/mm$^2$. The surface density of defects was also reflected in the surface roughness, which increased linearly with the density of defects.

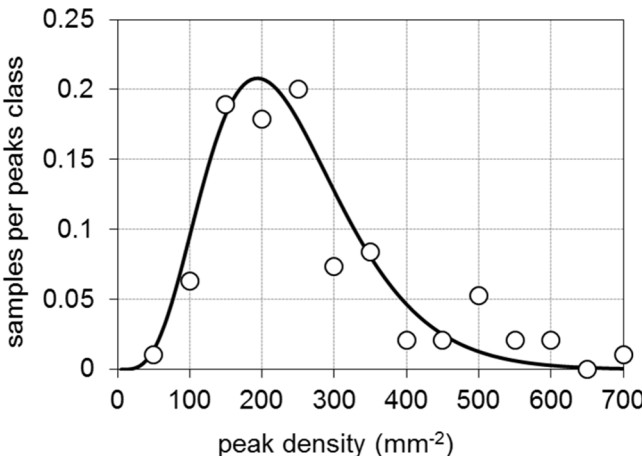

**Figure 13.** Peak (i.e., nodular defect) density distribution for TiN hard coatings prepared by low-voltage electron beam evaporation (BAI).

As can be seen from Figures 3 and 4, the surface densities of growth defects in the CC7 and AIP coatings were much higher than in the BAI coating, which was reflected in a greater surface roughness. In order to determine how the layer roughness affected its tribological properties, we performed tribological tests on a large number of samples prepared in different production batches. The wear rate measurements for BAI, CC7 and AIP TiN coatings with different levels of surface roughness were performed in nitrogen, oxygen and ambient air atmospheres. The results of the measurements are presented in Figure 14. The measurements clearly showed that the wear coefficient increased with the surface roughness of the coating and the most for the AIP coating. The wear rates in an

oxygen atmosphere were much larger (by one order of magnitude), compared with the rates observed in a nitrogen atmosphere. The values of the wear rates for the coatings tested in the ambient air were also smaller than those in the oxygen atmosphere, but still much higher than those measured in the nitrogen atmosphere. The difference between the wear coefficients in the oxygen and ambient air atmospheres could be explained by the influence of humidity in the ambient air [53].

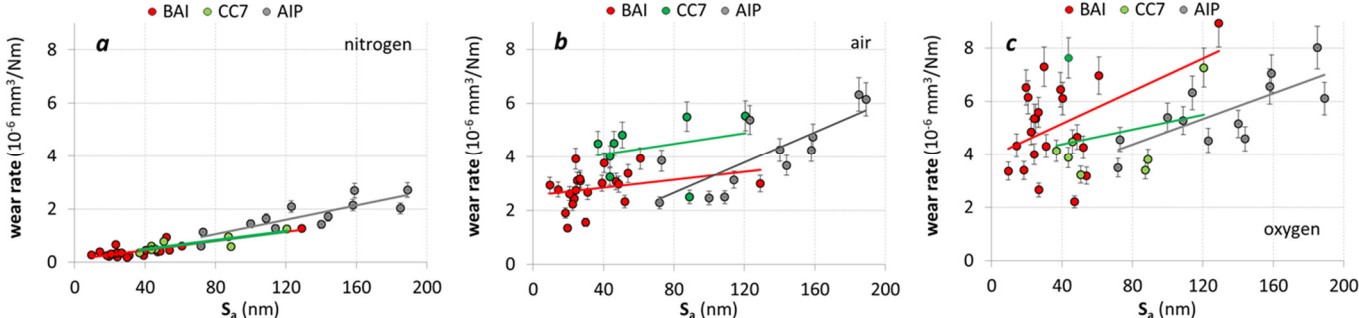

**Figure 14.** Wear rates of BAI, CC7 and AIP coatings as a function of surface roughness $S_a$. The wear tests were performed in nitrogen (**a**), ambient air (**b**) and oxygen (**c**) atmospheres.

The processes that take place in the wear track during the repeated sliding tests include: the breaking of nodular defects, coating surface smoothening, oxidation, formation of oxide patches, accumulation of wear debris at the track end and edges, and formation of roll-like debris (Figure 15). Previously, we explained that the formation of roll-like debris occurs due to plowing of the counter-body surface irregularities over the surface of the wear track [51]. Due to the large difference in the oxide layer molar volume in comparison with the molar volume of the nitride layer, the cracks formed in the oxide layer, which was mostly delaminated from the substrate. The dark regions on the backscattered-electron (BSE) image were oxide residues (Figure 15b). Oxides with lower atomic numbers than the coating appeared as dark areas, because they emit fewer back-scattered electrons in comparison to the coating material. We found that such oxide patches occurred preferentially at the sites of depressions and protrusions (at sites of carbide inclusions, nodular defects) and other surface irregularities (e.g., grooves, ridges).

Measurements on the alumina counterparts were also performed. 3D profile images of counterpart wear scars shown in Figure 16 reveal the existence of transfer layers on the ball sliding surface for all testing atmospheres. While the wear of the ball surface was similar for all three atmospheres, the amount of material transferred to the ball surface was the highest in the oxygen atmosphere and the lowest in nitrogen. As is evident from Figure 16, the lower wear of the coating in the nitrogen atmosphere was reflected in the lower transfer of material to the alumina ball. The tribofilm on the worn alumina surface was composed of the particles generated from both surfaces. Due to the high pressure and high temperature in the sliding contact, the wear particles were pressed together and sintered in the form of a tribofilm.

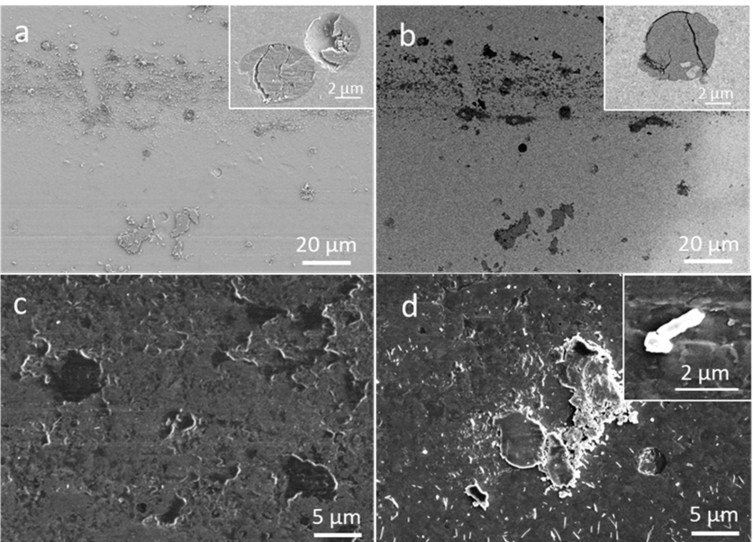

**Figure 15.** Secondary electron image (**a**) and backscattered electron image (**b**) of a wear track formed in BAI TiN coatings during a reciprocating sliding test performed in the ambient air; the anchoring points for wear debris and counter-body material transfer are depressions (see insets) as well as growth defects on the coating surface. SEM images of the wear track formed in AIP TiN coating during a reciprocating sliding test performed in the ambient air (**c**,**d**); the anchoring points for wear debris and counter-body material transfer are protrusions at the site of carbides (**c**), as well as broken nodular defects (**d**); roll-like debris are also visible in SEM image (**d**).

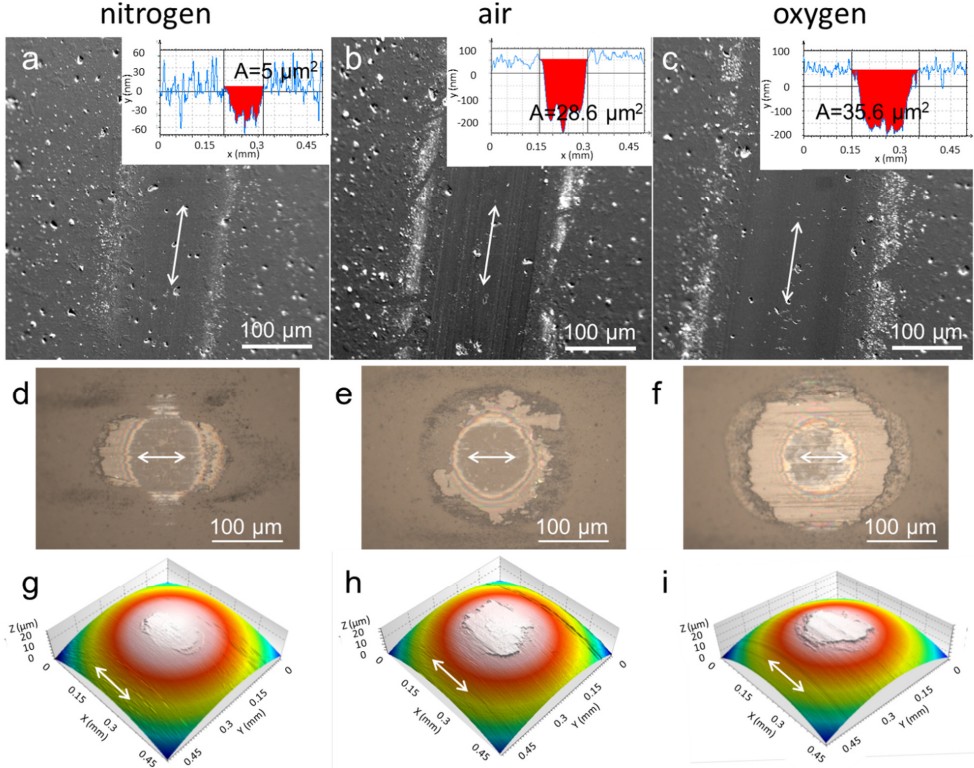

**Figure 16.** (**a**–**c**) SEM images of wear tracks formed in CC7 TiN coating during a reciprocating sliding test performed in nitrogen (**a**), ambient air (**b**), and oxygen (**c**) atmospheres. The cross-section areas (A) of corresponding wear tracks after 1000 cycles are shown in the insets. The areas of alumina ball wear scar are shown in the optical microscopy images (**d**–**f**) and 3D profilometer images (**g**–**i**). The images reveal the existence of transfer layers on the alumina ball for all testing atmospheres. Arrows indicate the sliding direction.

### 3. Conclusions

In this paper, we presented a comparative study of structural, microstructural, mechanical, topographical, and tribological properties of TiN coatings prepared by low-voltage electron beam evaporation, magnetron sputtering, and cathodic arc deposition. We tried to correlate the tribological behavior of these coatings with their intrinsic properties and friction conditions. We focused mainly on the investigation of tribological processes that take place in the initial phase of the sliding test. In this running-in period, the greatest impact was from different kinds of protrusions on the coating surfaces (e.g., nodular defects, droplets). Namely, all surface asperities caused abrasive wear to the softer counter-material surface and also the transfer of this material to the coating surface. Additionally, the protrusions (e.g., nodular defects) were crushed into small abrasive particles due to high pressure and shear forces during the sliding test. Therefore, we believe that in the running-in phase, nodular defects were the most intensive source of hard abrasive particles in the sliding contact. The longer running-in period and higher coefficient of friction of the coating prepared by the cathodic arc deposition were attributed to droplets on the surface of the as-deposited coating as well as those incorporated into the coating. Namely, relatively soft metal droplets increased the adhesion component of friction.

The negative impact of nodular defects on tribological performance can be reduced by post-polishing the as-deposited coating. Indeed, our tribological measurements showed that after post-polishing, the running-in period was shortened and the reduction in the coating wear rate was particularly enhanced.

In order to identify the influence of tribo-oxidation on friction and wear, the sliding tests on different types of TiN coatings were also conducted in different atmospheres (ambient air, nitrogen, oxygen). Oxygen promotes tribo-chemical reactions at the contact surface of the coating, while nitrogen suppresses them. This was particularly reflected in the coefficient of friction and wear, which were significantly higher in an oxygen atmosphere than in the nitrogen one. Oxides formed quickly in the sliding contact due to heating induced by friction. Due to the large difference in the oxide layer molar volume in comparison with the molar volume of the nitride layer, high compressive stresses appeared in the oxide layer. These stresses, together with high shear forces, caused the formation of cracks in the oxide film and finally its spallation. Only small patches of oxides in the wear track could be observed. Delaminated oxide fragments can act as abrasive particles, which cause an additional increase in the coefficient of friction and wear rate. In a test carried out in a nitrogen atmosphere, the formation of an oxide layer was avoided. Therefore, a more stable sliding contact between the two mating surfaces was formed. The more stable friction conditions were reflected in less fluctuation of the friction coefficient and, therefore, smoother friction curves. We found that the wear rate increased with the surface roughness of the coating. The wear rate in an oxygen atmosphere was much larger compared with that in a nitrogen atmosphere.

**Author Contributions:** Design of experiments, tribological and X-ray measurements, 3D profilometry, interpretation of experimental results, manuscript writing, P.P.; SEM and FIB analysis, manuscript review, A.D.; interpretation of tribological results, P.T. and A.M.; manuscript review, M.Č. and M.P. All authors have read and agreed to the published version of the manuscript.

**Funding:** This work was supported by Slovenian Research Agency (program P2-0082). We also acknowledge funding from the European Regional Development Funds (CENN Nanocenter, OP13.1.1.2.02.006).

**Institutional Review Board Statement:** Not applicable.

**Informed Consent Statement:** Not applicable.

**Data Availability Statement:** Not applicable.

**Acknowledgments:** The authors would also like to thank Jožko Fišer for performing some laboratory tests.

**Conflicts of Interest:** The authors declare no conflict of interest.

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
