# Peer review of "Comparative Study of Tribological Behavior of TiN Hard Coatings Deposited by Various PVD Deposition Techniques"

_coatings, doi:10.3390/coatings12030294_

Round 1

Reviewer 1 Report

Dear editor,

I have finished the reviewing process of the manuscript “coatings-1585457” intitled “Comparative study of tribological behavior of TiN hard coatings deposited by various PVD deposition techniques”. Although the paper is full of redundant information (authors tried to give insights and basic about different behaviors, e.g. theory of films growth, fracture toughness, changes on the preferential orientation, running in, stead state stages, etc, in the end the authors had a lot of work and somehow were able to explain the more important results of the manuscript. I recommend the publication of the manuscript after the following changes. I require the editor intervention in point i) .

  1. i) I can understand that explanation of basic about films growth structure, tribology, etc can indeed give a more clear understanding to readers which are in the field at few time. However, in manuscripts we normally discuss the results supported by references,howeverm basics principles are rarely given high quality papers. In my opinion, this needs polishing in this manuscript. I leave this decision to the editor decide and ask the authors to change that according with my comment.
  2. ii) abstract should be improved.
  • In first paragrapher authors stated that the objective of the paper is to “The objective of the present work was to investigate and compare the microstructure, 11 crystal structure, surface topography and mechanical properties of TiN hard coatings deposited in 12 industrial PVD deposition systems using three deposition methods”, but immediately below they say “The main goal of this work, 14 however, was to find a relation between these coating properties and their tribological behavior”. The authors should clear sate what is the objective of the manuscript.
  • The authors stated “The tribological characteristics of all coatings in dry sliding contact with an alumina ball as a 16 counter material were evaluated at room temperature by ball-on-disc test with reciprocal motion.”, however, immediately below they say “Tribological interactions between the two mating surfaces significantly changed 24 after polishing the as-deposited coating surface. The sliding tests on different types of TiN coatings 25 were also conducted in different atmospheres (air, nitrogen, oxygen) in order to find the influence 26 of tribo-oxidation on friction and wear.” The authors should organize their ideas in such way this inconsistences are avoided. There are some more examples along the manuscript, please correct it.

iii) number of keywords should be reduced. Normally journals asks a maximum of 5 keywords.

iv)In general authors don’t compare their results with the literature. This is a critical issue missing in the manuscript. Are the authors coatings behaving better than the TiN coatings published in the literature?

  1. v) Chemical composition of coatings is missing in the literature. Are the coatings stoichiometric? If not it can influence the hardness values and consequently tribological performance?
  2. vi) explanation of the differences in microstructure is not easy to follow. Authors should be more clear on the ideas and go more direct to the point.

vii) Fig 6 should be changed. Color caption and figure caption are not correct.

viii) First paragrapher of sub-section 3.4 is not correct. Is not the hardness and E that depends of tribology is the opposite.

  1. ix) title of sub-section 3.5 could be shorten

Author Response

We wish to express our appreciation for your comments, which have helped us improve the paper.

NOTE : black color: reviewer’s comments, blue color: authors’ responses and highlighted in the manuscript with red color.

Reviewer’s comments and authors’ responses

Referee remark 1: I can understand that explanation of basic about films growth structure, tribology, etc can indeed give a more clear understanding to readers which are in the field at few time. However, in manuscripts we normally discuss the results supported by references,howeverm basics principles are rarely given high quality papers. In my opinion, this needs polishing in this manuscript. I leave this decision to the editor decide and ask the authors to change that according with my comment.

We considered this remark. The manuscript are rearranged and the results are presented more transparent. We removed some superfluous sentences in introduction. New references were added.

Referee remark 2: abstract should be improved.

We considered this remark and we rearranged the abstract.

Referee remark 3: In first paragrapher authors stated that the objective of the paper is to “The objective of the present work was to investigate and compare the microstructure, 11 crystal structure, surface topography and mechanical properties of TiN hard coatings deposited in 12 industrial PVD deposition systems using three deposition methods”, but immediately below they say “The main goal of this work, 14 however, was to find a relation between these coating properties and their tribological behavior”. The authors should clear sate what is the objective of the manuscript.

We considered the reviewer's remark. Therefore, we wrote more explicitly what the main objectives of this study were (see abstract and the last paragraph of the introduction).

Referee remark 4: The authors stated “The tribological characteristics of all coatings in dry sliding contact with an alumina ball as a 16 counter material were evaluated at room temperature by ball-on-disc test with reciprocal motion.”, however, immediately below they say “Tribological interactions between the two mating surfaces significantly changed 24 after polishing the as-deposited coating surface. The sliding tests on different types of TiN coatings were also conducted in different atmospheres (air, nitrogen, oxygen) in order to find the influence of tribo-oxidation on friction and wear.” The authors should organize their ideas in such way this inconsistences are avoided. There are some more examples along the manuscript, please correct it.

We considered the reviewer's remark. We divided the results of tribological test into two sections. In the first one we presented the results related to the influence of coating topography on friction and wear. In the second one we descried the influence of  different surrounding atmospheres on the friction and wear.the results and added some new references.

Referee remark 5: ) number of keywords should be reduced. Normally journals asks a maximum of 5 keywords.

We reduced the number of keywords

Referee remark 6: In general authors don’t compare their results with the literature. This is a critical issue missing in the manuscript. Are the authors coatings behaving better than the TiN coatings published in the literature?

We considered the reviewer's remark as far as possible. Namely, it is difficult to compare our tribological results with those in the literature, because the sliding couples and friction conditions are very different. We limited ourselves to the investigation of tribological processes that take place in the initial phase of the sliding test (the first 1000 cycles). However only a few papers exist, in which the authors discuss the running-in period of friction process, but these refer to other types of coatings, e.g. TiAlN / VN (Luo Q.,  Tribo. Letters, 2010, 37, 529-539) or nc-TaC/aC (Olofsson et al., Wear, 2011, 271, 2046-2057). To the best of our knowledge, the only article in which the authors discussed the tribological behavior of TiN in the initial phase is "P. Harlin et al., Surf. Coat. Technol., 2006, 201 (7), 4253". In this paper, the influence of coating surface roughness on the friction and wear behavior of ball bearing steel sliding against TiN- and WC/C coated high speed steel under dry sliding contact conditions were evaluated. Both coatings were deposited on a PM HSS substrates, that were grinded or polished to different surface roughnesses (from 59 to 185 nm). After coating deposition some of the coated samples were post-polished in order to remove protruding coating surface asperities (macro particles) from the coating surface.

 All three above mention papers are added to the references.

Referee remark 7: Chemical composition of coatings is missing in the literature. Are the coatings stoichiometric? If not it can influence the hardness values and consequently tribological performance?

We agree that the variation of the stoichiometry of TiN coatings could creates differences in their mechanical properties. However, in our case we performed EDX analysis and the results shown that all type of coatings were stoichiometric. An additional proof that all our TiN coating have a stoichiometric composition is their distinctly golden color. When TiN coating is understoichiometric (x<l), then its color is silver, while the cooper color is characteristic for an overstoichiometric (x > 1) composition. In the first sentence of chapter 2 we added all TiN coatings had a stoichiometric composition.

Referee remark 8: explanation of the differences in microstructure is not easy to follow. Authors should be more clear on the ideas and go more direct to the point.

In order to improve the comprehensibility and transparency of the text, we have eliminated some superfluous sentences from this chapter.

Referee remark 9: Fig 6 should be changed. Color caption and figure caption are not correct

Corrected

Referee remark 10: First paragrapher of sub-section 3.4 is not correct. Is not the hardness and E that depends of tribology is the opposite.

We considered the reviewer's remark

Referee remark 11: title of sub-section 3.5 could be shorten

Corrected

Referee remark 12: English language and style are fine/minor spell check required

The revised manuscript was grammatically checked by an English language editing services.

Reviewer 2 Report

In the current research work, the authors investigated the tribological behavior of three different TiN coatings (produced in industrial deposition systems by means of electron beam evaporation, unbalanced magnetron sputtering and arc evaporation) in the atmospheres of nitrogen, ambient air, and oxygen. This is a fully comprehensive study, and the manuscript is well structured and written. I certainly recommend it for publication. Below are a few typos I have stumbled upon while reading the manuscript and a few minor comments:

Abstract. Consider expanding on the novelty of the study (i.e., emphasize the challenges and questions that remained open in understanding the tribological behavior of TiN hard coatings) as well as on the implications of the findings in a few additional sentences.

Introduction. This section is unnecessarily lengthy and should be shortened. For example, it is quite superfluous to explain why TiN has a golden color and describe all possible applications in detail. Page 3 can easily be boiled down to 1-2 sentences.

Conclusions. In the last paragraph, consider expanding oh how different deposition techniques affected the results.

Lines 35-36: “The first TiN hard coating was deposited on cemented carbide inserts in 1969 using the chemical vapour deposition (CVD) process” – A reference is needed here. If Ref. 1 is meant, it is not TiN but TiC that was deposited on cemented carbide inserts in 1969 by CVD, cf. Ref. 1: “Tribological coatings as we know them today really received their start in 1969 with the introduction of titanium carbide (TiC) coated cemented carbide cutting tool inserts by Sandvik and Krupp-Widia that were deposited using the high temperature chemical vapor deposition process”.

Line 68: “about years later” – A missing number?

Line 86: “bellow 600 °C” – below.

Line 178: “Prior the coating deposition” – Prior to.

Line 283: “(200) plan is” – plane.

Fig.6 – Consider sorting the names in the legend according to the order of the XRD patterns, i.e. CC7, BAI, and AIP. Although I would suggest to sort both the XRD patterns as well as the names in order BAI, CC7, and AIP, to be consistent throughout the manuscript.

Line 443: “Green et al.” – Greene et al.

Table 3 – Consider adding the errors for H/E and H3/E2 (i.e., propagation of uncertainty).

Line 488: “performed on in flat areas” – performed on flat areas.

Line 500: “wear coefficient (see Fig. 11)” – Consider indicating “(chapter 3.5)” as a cross-reference since Figs. 8-10 haven’t been mentioned so far.

Line 522: “to about 0.35 for the AIP coating” – Isn’t it rather 0.45?

Fig. 7 – (a), (b), (c) are missing. Also, consider expanding on the figure caption.

Fig. 9 – COFs obtained in a nitrogen atmosphere (red) in (a) and (c) are different from Fig.7. Were some new tests carried out?

Line 632: “while they are the lowest in a nitrogen atmosphere” – COF is the lowest in ambient air, e.g. Fig. 11.

Line 738: “The tribofilm … is composed from” – Composed of.

Lines 748-750: “tribological behaviour of TiN hard coatings … and its mechanical properties” – coatings/their or coating/its.

Lines 783-784: “which are higher for …, significantly than in” – significantly higher than?

Author Response

We wish to express our appreciation for your comments, which have helped us improve the paper.

NOTE : black color: reviewer’s comments, blue color: authors’ responses and highlighted in the manuscript with red color.

Reviewer’s comments and authors’ responses

Referee remark 1: Abstract. Consider expanding on the novelty of the study (i.e., emphasize the challenges and questions that remained open in understanding the tribological behavior of TiN hard coatings) as well as on the implications of the findings in a few additional sentences.

We rearranged the abstract according to the reviewer’s suggestion.

Referee remark 2: Introduction. This section is unnecessarily lengthy and should be shortened. For example, it is quite superfluous to explain why TiN has a golden color and describe all possible applications in detail. Page 3 can easily be boiled down to 1-2 sentences.

The introduction is shortened. Some superfluous sentences are eliminated.

Referee remark 3: Conclusions. In the last paragraph, consider expanding oh how different deposition techniques affected the results.

We considered this remark and we rearranged the conclusions and presented them more precisely.

Referee remark 4: Lines 35-36: “The first TiN hard coating was deposited on cemented carbide inserts in 1969 using the chemical vapour deposition (CVD) process” – A reference is needed here. If Ref. 1 is meant, it is not TiN but TiC that was deposited on cemented carbide inserts in 1969 by CVD, cf. Ref. 1: “Tribological coatings as we know them today really received their start in 1969 with the introduction of titanium carbide (TiC) coated cemented carbide cutting tool inserts by Sandvik and Krupp-Widia that were deposited using the high temperature chemical vapor deposition process”.

The reviewer is right. Sandvik Coromant and Krupp-Widia introduced the CVD TiC coating in 1969. The CVD TiN coating appeared on market just a little later (1970). We have corrected the text and added the corresponding reference ([2] Feinberg B., Longer life from TiN tools, Mfg. Eng. Manag., 1971, vol 67, no.1, 6-18).

Referee remark 5: Line 68: “about years later” – A missing number?

“about years later”  was replaced with »about 20 years later«

Referee remark 6: Line 86: “bellow 600 °C” – below.

corrected

Referee remark 7: Line 178: “Prior the coating deposition” – Prior to.

corrected

Referee remark 8: Line 283: “(200) plan is” – plane.

corrected

Referee remark 9: Fig.6 – Consider sorting the names in the legend according to the order of the XRD patterns, i.e. CC7, BAI, and AIP. Although I would suggest to sort both the XRD patterns as well as the names in order BAI, CC7, and AIP, to be consistent throughout the manuscript.

corrected according to the reviewer’s suggestion

Referee remark 10: Line 443: “Green et al.” – Greene et al.

corrected

Referee remark 11: Table 3 – Consider adding the errors for H/E and H3/E2 (i.e., propagation of uncertainty).

added

Referee remark 12: Line 488: “performed on in flat areas” – performed on flat areas.

corrected

Referee remark 13: Line 500: “wear coefficient (see Fig. 11)” – Consider indicating “(chapter 3.5)” as a cross-reference since Figs. 8-10 haven’t been mentioned so far.

corrected

Referee remark 14: Line 522: “to about 0.35 for the AIP coating” – Isn’t it rather 0.45?

You are right. I'm sorry for the mistake.

Referee remark 15: Fig. 7 – (a), (b), (c) are missing. Also, consider expanding on the figure caption.

the designations was added

Referee remark 16: Fig. 9 – COFs obtained in a nitrogen atmosphere (red) in (a) and (c) are different from Fig.7. Were some new tests carried out?

You are right. However, we have to consider that the measurements in Figures 7 and 9 were performed on samples from different batches. Let me explain more precisely. Both measurements in Figures 7a were made on the same sample. Also, all three measurements given in Figures 9a were performed on the same sample but from a different batch. As we explained at the end of Chapter 3.5, coatings from different batches (and even from the same batch) differ from each other in the density of surface defects. Therefore, the measured friction curves also differ slightly from each other.

Referee remark 17: Line 632: “while they are the lowest in a nitrogen atmosphere” – COF is the lowest in ambient air, e.g. Fig. 11.

the sentence was corrected.

Referee remark 18: Line 738: “The tribofilm … is composed from” – Composed of.

corrected

Referee remark 19: Lines 748-750: “tribological behaviour of TiN hard coatings … and its mechanical properties” – coatings/their or coating/its.

Corrected

Referee remark 20: Lines 783-784: “which are higher for …, significantly than in” – significantly higher than?

Corrected

Referee remark 21: English language and style are fine/minor spell check requied

The revised manuscript was grammatically checked by an English language editing services.

Round 2

Reviewer 1 Report

Dear editor, the authors sucessefully adressed all my questions and changed the manuscript accordingly. The manuscript is now in condition for publication. 

Best Regards